# Chronic diseases and multimorbidity among unemployed and employed persons in the Netherlands: a register-based cross-sectional study

Berivan Yildiz [1], Merel Schuring,[1] Marike G Knoef,[2] Alex Burdorf [1]

¹Department of Public Health, Erasmus University Medical Center, Rotterdam, The Netherlands
²Department of Economics, Leiden University, Leiden, Zuid-Holland, The Netherlands

**Correspondence to**
Dr Merel Schuring;
m.schuring@erasmusmc.nl

## ABSTRACT

**Objectives** The first objective of this study was to describe the age-specific prevalence of chronic diseases and multimorbidity among unemployed and employed persons. The second objective was to examine associations of employment status and sociodemographic characteristics with chronic diseases and multimorbidity.

**Design** Data linkage of cross-sectional nationwide registries on employment status, medication use and sociodemographic characteristics in 2016 was applied.

**Setting** Register-based data covering residents in the Netherlands.

**Participants** 5 074 227 persons aged 18–65 years were selected with information on employment status, medication use and sociodemographic characteristics.

**Outcome measures** Multiple logistic regression analysis and descriptive statistics were performed to examine associations of employment and sociodemographic characteristics with the prevalence of chronic diseases and multimorbidity. The age-specific prevalence of six common chronic diseases was described, and Venn diagrams were applied for multimorbidity among unemployed and employed persons.

**Results** Unemployed persons had a higher prevalence of psychological disorders (18.3% vs 5.4%), cardiovascular diseases (20.1% vs 8.9%), inflammatory diseases (24.5% vs 15.8%) and respiratory diseases (11.7% vs 6.5%) than employed persons. Unemployed persons were more likely to have one (OR 1.30 (1.29–1.31)), two (OR 1.74 (1.73–1.76)) and at least three chronic diseases (OR 2.59 (2.56–2.61)) than employed persons. Among unemployed persons, psychological disorders and inflammatory conditions increased with age but declined from middle age onwards, whereas a slight increase was observed among employed persons. Older persons, women, lower educated persons and migrants were more likely to have chronic diseases.

**Conclusion** Large differences exist in the prevalence of chronic diseases and multimorbidity among unemployed and employed persons. The age-specific prevalence follows a different pattern among employed and unemployed persons, with a relatively high prevalence of psychological disorders and inflammatory conditions among middle-aged unemployed persons. Policy measures should focus more on promoting employment among unemployed persons with chronic diseases.

## Strengths and limitations of this study

► This is the first study that describes the prevalence of chronic diseases and multimorbidity among unemployed and employed persons, using objective register-based data rather than self-reported health outcomes.

► A strength of this study is applying data linkage of nationwide registries that capture the whole population, facilitating precise estimations of associations between health and employment, and offering us the possibility to investigate specific subgroups (age-specific prevalence).

► Except for back pain and musculoskeletal disorders, this study investigated a broad range of chronic diseases such as cardiovascular diseases, psychological disorders, diabetes and respiratory diseases.

► Causal effects of having a chronic disease on employment status, or vice versa cannot be distinguished because of the use of cross-sectional data and the bidirectional nature of health and employment.

## INTRODUCTION

The relationship between unemployment and health has been well established.[1 2] In general, unemployed individuals have worse mental and physical health compared with employed individuals.[2–5]

These health inequalities between employed and unemployed persons can be explained by two different hypotheses. First, according to the causation hypothesis, persons who become unemployed will deteriorate in health, whereas unemployed persons who enter paid employment will improve in health.[6] Second, the selection hypothesis describes that persons who leave paid employment already have lower levels of health before leaving employment compared with those who remain employed, whereas persons who enter paid employment already have a better health status before entering

employment compared with persons who remain unemployed.[7]

Chronic diseases can affect an individual's employment status due to experienced functional limitations and a poor-quality of life.[8] Two studies have shown that long-term health conditions such as cardiovascular diseases and diabetes were associated with unemployment.[9 10] In addition, a recent systematic review provided evidence that individuals with diabetes were more likely to be out of the labour force.[9] However, studies investigating chronic diseases and employment have mostly focused on a single disease, whereas many persons with a chronic disease are likely to suffer from multiple chronic diseases, especially among older age groups. There is increasing evidence that persons with multimorbidity—the co-occurrence of at least two chronic diseases within an individual—may be more likely to have poorer functional outcomes and thus may be also more often out of the labour market than those with a single chronic disease or no chronic disease. For instance, a study among Australian workers with multiple health problems showed that individuals with four or more health problems were far less likely to be employed compared with those with no health condition.[11] Another study among persons with back complaints found that the co-occurrence of cardiovascular diseases resulted in a 10-fold increased risk of unemployment compared with those with back complaints alone.[12]

So far, findings on the association of chronic diseases and multimorbidity with unemployment have been based mostly on self-reported health outcomes. Self-reported health outcomes are known to be vulnerable to reporting bias and justification bias. Therefore, a more objective approach is preferred in order to make more precise estimations of the prevalence of diseases. One way to objectively investigate the presence of chronic diseases is by using pharmacy data.[13] Pharmacy data provide a reliable information source and often cover a large population.[14] Administrative databases such as drug prescription can be used to identify persons with chronic diseases. So far, only a few studies have used register-based data in order to investigate chronic diseases among unemployed persons. A Danish register-based study found a higher prevalence of mental disorders and cardiovascular disorders among unemployed persons receiving social benefits compared with employed persons.[15] In line with this, another register-based study showed that long-term unemployment was associated with a higher risk of antidepressant use.[16]

To our knowledge, none of the evidence on the association between multimorbidity and employment status has been based on register-based data. Therefore, the present study aimed to investigate the prevalence of chronic diseases and multimorbidity among unemployed and employed persons. The large register data enabled to investigate specific subgroups (eg, age-specific prevalences and associations). The second aim was to examine associations of employment status and sociodemographic characteristics with chronic diseases and multimorbidity.

Nation-wide data from the Netherlands on drug prescription, employment status and sociodemographic characteristics in 2016 were used.

## METHODS

### Study population and design

Register data covering information on all Dutch residents in 2016 were used. Statistics Netherlands provided individual-level databases on demographics, education, labour market status and prescribed medication. All Dutch residents were pseudonymised using a personal unique number. Data registries were linked at the individual level using these pseudonymised numbers. No informed consent was needed for this study since authorised research institutes in the Netherlands are by law allowed to use pseudonymised register-based data for research purposes.

Individuals aged between 18 and 64 years with available information on employment status were selected (n=10 514 271). This selection captured individuals who were not eligible for exit from paid employment through statutory national retirement schemes. Due to the lack of nation-wide education registers in the past, many older persons had missing data on educational level. Also, the current register only includes formal education obtained at institutes financed by government. Therefore, we excluded 31.9% of individuals with missing data on educational level (n=3 356 002). Within the population with available data on all sociodemographic characteristics (n=7 158 269), 4 566 644 persons were classified as employed and 507 583 persons were classified as unemployed. In total, 5 074 227 subjects were included in the present study.

### Employment status

The database on social economic category per month provided information on employment status of participants for each month in the year 2016. The main source of income for each consecutive month was used to classify persons as employed or unemployed. Individuals who were in paid employment or self-employed for at least 9 months in 2016 were classified as employed (n=4 566 644). Individuals who were out of the labour market and received either social benefits or unemployment benefits for at least 9 months were classified as unemployed (n=507 583).

### Chronic diseases and multimorbidity

The database on medication use in 2016 (Medicijntab) provides information on purchased drugs that were reimbursed by the healthcare insurances. The drugs were identified using the WHO Anatomical Therapeutic Chemical (ATC) classification codes.[17] In line with the study of Huber et al, specific chronic diseases were identified based on these ATC-codes.[13] For instance, psychological disorders were identified by ATC-codes that were assigned to drugs such as antidepressants and anxiolytics, whereas inflammatory conditions were identified

by the ATC-code that was assigned to non-steroidal anti-inflammatory drugs (online supplementary table 1).

The presence of a specific chronic disease was dichotomised into having or not having a chronic disease. Multimorbidity was investigated as (1) the number of chronic diseases and (2) the combinations of four common chronic diseases with the highest prevalence in the study population. For the first approach of multimorbidity, the total number of chronic diseases was computed for each participant, based on 21 different chronic diseases that could be identified by medication prescription.[13] This measure of multimorbidity was categorised into four groups: no chronic disease, one chronic disease, two chronic diseases and at least three chronic diseases. For the second approach, we used the following four chronic diseases to describe their co-occurrence: cardiovascular diseases, psychological disorders, inflammatory conditions and respiratory diseases.

### Sociodemographic variables

The databases on sociodemographic characteristics provide information on age, gender, education and migration background. A dichotomous variable was computed for employment status (employed vs unemployed). Educational level was categorised into three educational groups: high (higher vocational training or university), intermediate (higher secondary and intermediate vocational training) and low education (pre-primary education, primary education and lower secondary education). Age was categorised into four age groups (18–30, 30–45, 45–55, 55–65). Migration background was categorised as native Dutch, Moroccan, Turkish, Surinamese and Antillean, other Western, and other non-Western.

### Analyses

Descriptive statistics were used to describe the prevalence of chronic diseases and multimorbidity among employed and unemployed persons. The association of sociodemographic characteristics (age, sex, education and migration background) and employment status with (multiple) chronic diseases was examined using multiple logistic regression analysis in the total study population (employed and unemployed persons). Separate logistic regression analyses were done for each number of chronic diseases (dependent variable): (1) one chronic disease (2) two chronic diseases and (3) three or more chronic diseases. In these analyses, having no chronic diseases was used as the reference category. Logistic regression analyses were adjusted for age, sex, educational level and migration background. The association between employment status and multimorbidity stratified by age was also investigated. To test for possible selection bias, sensitivity analyses were performed by including individuals with missing data on educational level.

The prevalence of multimorbidity was also described as the proportion of individuals with all potential combinations of four exclusive chronic diseases. All combinations of co-occurrence between the four chronic diseases were presented in a Venn diagram for employed and unemployed persons. The age-specific prevalence of these four chronic diseases among unemployed and employed persons was presented. In order to distinguish specific conditions within a chronic disease, the age-specific prevalence of specific medicines was also investigated for cardiovascular diseases and psychological disorders. The latter was not investigated for the other chronic diseases because less specific medicines could be distinguished within the other chronic disease.

### Patient and public involvement

No patients were involved.

## RESULTS

Unemployed persons were more often older than 45 years (58.2%), female (54.5%), lower educated (51.8%) and from non-Dutch origin (48.1%) compared with employed persons (19.4%). Differences in the prevalence of chronic diseases between unemployed and employed persons were highest for psychological disorders. Compared with employed persons, unemployed persons had a higher prevalence of psychological disorders (18.3% vs 5.4%), cardiovascular diseases (20.1% vs 8.9%), inflammatory conditions (24.5% vs 15.8%), psychotic illness (6.2% vs 0.8%), respiratory diseases (11.7% vs 6.5%) and diabetes (7.2% vs 2.0%). (table 1)

The prevalence of multimorbidity was also higher for unemployed persons compared with employed persons. The co-occurrence of both psychological disorders and inflammatory conditions was higher among unemployed persons (3.6%+0.9%+1.2%+0.5%=6.2%) than among employed persons (1.0%+0.2%+0.2%+0.1%=1.5%). In addition, the co-occurrence of cardiovascular diseases and inflammatory conditions was higher among unemployed persons (5.5%) compared with employed persons (2.1%). The prevalence of having both cardiovascular diseases and psychological disorders was 4.9% among unemployed persons compared with 0.9% among employed persons (figure 1, online supplementary table 2).

At all ages, unemployed individuals had a higher prevalence of all four chronic diseases compared with employed individuals. The prevalence of psychological disorders increased with age followed by a decrease from middle age onwards among unemployed persons, whereas a slight increase was observed among employed persons. The same pattern was observed for inflammatory conditions. The prevalence of cardiovascular diseases and respiratory diseases increased with age among both unemployed and employed persons (figure 2).

Among unemployed persons, the use of antidepressants, anxiolytics and, hypnotics and sedatives was highest at middle age. The prevalence of antidepressants was higher than the use of anxiolytics, hypnotics and sedatives among unemployed persons for all age groups. The use of antithrombotic and cardiac agents, beta blockers

**Table 1** Characteristics of the study population by employment status

| Age | Unemployed (n=507 583) | Employed (n=4 566 644) |
|---|---|---|
| | n (%) | n (%) |
| 18–30 | 54 807 (10.8) | 1 223 211 (26.8) |
| 30–45 | 157 238 (31.0) | 1 808 274 (39.6) |
| 45–55 | 147 865 (29.1) | 1 033 090 (22.6) |
| 55–65 | 147 673 (29.1) | 502 069 (11.0) |
| **Sex** | | |
| Male | 230 856 (45.4) | 2 405 740 (52.7) |
| Female | 276 727 (54.5) | 2 160 904 (47.3) |
| **Educational level** | | |
| High | 73 893 (14.6) | 2 022 717 (44.3) |
| Middle | 170 857 (33.7) | 1 906 830 (41.8) |
| Low | 262 833 (51.8) | 637 097 (14.0) |
| **Migration background** | | |
| Native Dutch | 263 196 (51.9) | 3 680 071 (80.6) |
| Moroccan | 35 441 (7.0) | 76 841 (1.7) |
| Turkish | 27 131 (5.3) | 97 716 (2.1) |
| Surinamese and Antillean | 40 446 (8.0) | 145 478 (2.3) |
| Other Western | 37 420 (7.4) | 311 005 (6.8) |
| Other non-Western | 103 949 (20.5) | 255 533 (5.6) |
| **Chronic diseases** | | |
| Inflammatory conditions | 124 411 (24.5) | 721 304 (15.8) |
| Cardiovascular diseases | 101 917 (20.1) | 405 200 (8.9) |
| Psychological disorders | 92 956 (18.3) | 248 520 (5.4) |
| Respiratory diseases | 59 557 (11.7) | 296 817 (6.5) |
| Diabetes | 36 662 (7.2) | 89 382 (2.0) |
| Psychotic illness | 31 308 (6.2) | 34 377 (0.8) |
| **No of chronic diseases** | | |
| 0 | 193 412 (38.1) | 2 877 313 (63.0) |
| 1 | 118 688 (23.4) | 1 007 275 (22.1) |
| 2 | 79 719 (15.7) | 394 396 (8.6) |
| ≥3 | 115 764 (22.8) | 287 660 (6.3) |

and ACE inhibitors, and diuretics and calcium-channel blockers was highest among older unemployed and employed persons. The prevalence of beta blockers and ACE inhibitors, and antithrombotic agents was higher than diuretics and calcium-channel blockers, and cardiac agents among unemployed persons of all age groups (online supplementary figure 1).

Unemployed persons were more likely to have (multiple) chronic diseases compared with employed persons. Unemployed persons were more likely to have one (OR 1.30, 95% CI 1.29 to 1.31), two (OR 1.74, 95% CI 1.73 to 1.76) and at least three chronic diseases (OR 2.59, 95% CI 2.56 to 2.61) than employed persons. Among unemployed persons, 23% had at least three chronic diseases compared with 6% among employed persons. Women (OR 1.49, 95% CI 1.49 to 1.50), older individuals (OR 1.32 to 2.27), middle and low educated persons (OR 1.32 to 1.49) and non-western migrants (OR 1.01 to 1.34) were also more likely to have a chronic disease. In addition, women (OR 1.61, 95% CI 1.60 to 1.63), older persons (OR 2.72 to 17.08), lower educated persons (OR 2.11 to 3.32) and non-western migrants (OR 1.02 to 2.03) were also more likely to have multiple (≥3) chronic diseases (table 2). Comparable results were found in sensitivity analyses on the total study population with inclusion of individuals who had missing data (n=7 576 196) on educational level.

Within all age groups, unemployed persons had a higher risk of having one, two and at least three chronic diseases than employed persons. Especially, among persons aged 18–30 years, unemployed persons were more likely to have one (OR 1.65, 95% CI 1.61 to 1.69), two (OR 2.84, 95% CI 2.75 to 2.93) and at least three (OR 5.20, 95% CI 4.99 to 5.42) chronic diseases compared with employed persons. These effect estimates were lower among older age groups (online supplementary table 3).

## DISCUSSION

In this large register-based study, unemployed persons had a higher prevalence of cardiovascular diseases, psychological disorders, inflammatory conditions, respiratory diseases and multimorbidity compared with employed persons. Between unemployed and employed persons, the largest differences were observed for cardiovascular diseases and psychological disorders. The prevalence of psychological disorders and inflammatory conditions was highest among unemployed persons in the middle age group. Women, older individuals, lower educated persons and non-western migrants were more likely to have one chronic disease as well as multiple chronic diseases. Among younger persons (18–30 years), a stronger association between chronic disease and unemployment was found compared with higher age groups.

The higher prevalence of chronic diseases among unemployed persons in the current study is in line with other studies showing that unemployed persons have a poorer mental and physical health status.[2 5 18] For instance, unemployed persons had high risks of common mental disorders such as depression.[19–21] Our study added to the current literature by comparing the age-specific prevalence of chronic diseases between unemployed and employed persons. Between unemployed and employed persons, the largest differences were observed for cardiovascular diseases and psychological disorders. Although the overall prevalence of cardiovascular diseases was much higher among unemployed persons,

**Figure 1** Multimorbidity of cardiovascular diseases (CVD), psychological disorders (PD), inflammatory conditions (IC) and respiratory diseases (RD) among unemployed (n=507 583) and employed (n=4 566 644) persons .

the age-specific patterns showed small differences, indicating that the higher age among unemployed persons was largely responsible for the higher prevalence of cardiovascular diseases.

A remarkable finding was the different pattern of the age-specific prevalence of psychological disorders between unemployed and employed persons. Among both employed and unemployed persons, the prevalence of psychological disorders increased with age. However, this increase was more profound among unemployed persons with a peak around middle age. This pattern among unemployed persons can be explained by studies arguing that before and after middle age, individuals tend to suffer less from unemployment compared with persons of middle age.[22] Persons of middle age often have family responsibilities, increasing the financial importance of a job, whereas younger and older persons experience less financial pressure and thus less psychological distress due to unemployment.[23] The other way around, it can also be that the combination of family responsibilities and work lead to pressure and cause both health problems as well as unemployment. Furthermore, it has also been hypothesised that persons of middle age are more likely to aim for a successful career which leads to employment being more important for their mental health than it is for older persons who are almost finishing their careers, and for younger persons who have recently entered paid employment.[24] The age-specific prevalence of antidepressants use, anxiolytics, and hypnotics and sedatives among unemployed persons confirms this theory by showing the highest prevalence at middle age and a lower prevalence at younger and older ages.

It was checked whether the decline in the prevalence of use of antidepressants, anxiolytics, hypnotics and sedatives after the age of 50 would be different among persons receiving a disability benefit. Namely, it might be possible that older unemployed persons with chronic diseases are more likely to receive disability benefits rather than unemployment or social benefits, and therefore the age-specific prevalence of these medicines declines among unemployed persons. However, also among persons with disability benefits, a decline was observed from middle age onwards for these medicines (results not shown).

Another interesting finding was an increase in the prevalence of inflammatory conditions with increasing age, followed by a decrease from middle age onwards among unemployed persons. This finding can be explained by the medicines NSAIDs that have been used to identify inflammatory conditions as a chronic disease in the present study. NSAIDs are pain killers with anti-inflammatory effects and are known to cause serious adverse effects.[25] Therefore, NSAIDs are cautiously or not prescribed among individuals aged 60 years or older, who suffer from cardiovascular diseases and already use other medicines for other (chronic) health conditions.[26] In the present study, the prevalence of cardiovascular diseases and multimorbidity was higher among unemployed persons compared with employed persons. Thus, it may be that NSAIDs are less prescribed among older persons because of multimorbidity with cardiovascular disorders and associated polypharmacy—the use of multiple medicines. Therefore, the higher prevalence of multimorbidity among unemployed persons may explain the decrease in the prevalence of inflammatory conditions with increasing age.

Unemployed persons had a higher prevalence of multimorbidity than employed persons. It is likely that the healthy worker selection process is more prominent among persons with multiple diseases than single diseases.[27] According to the causation mechanism, it

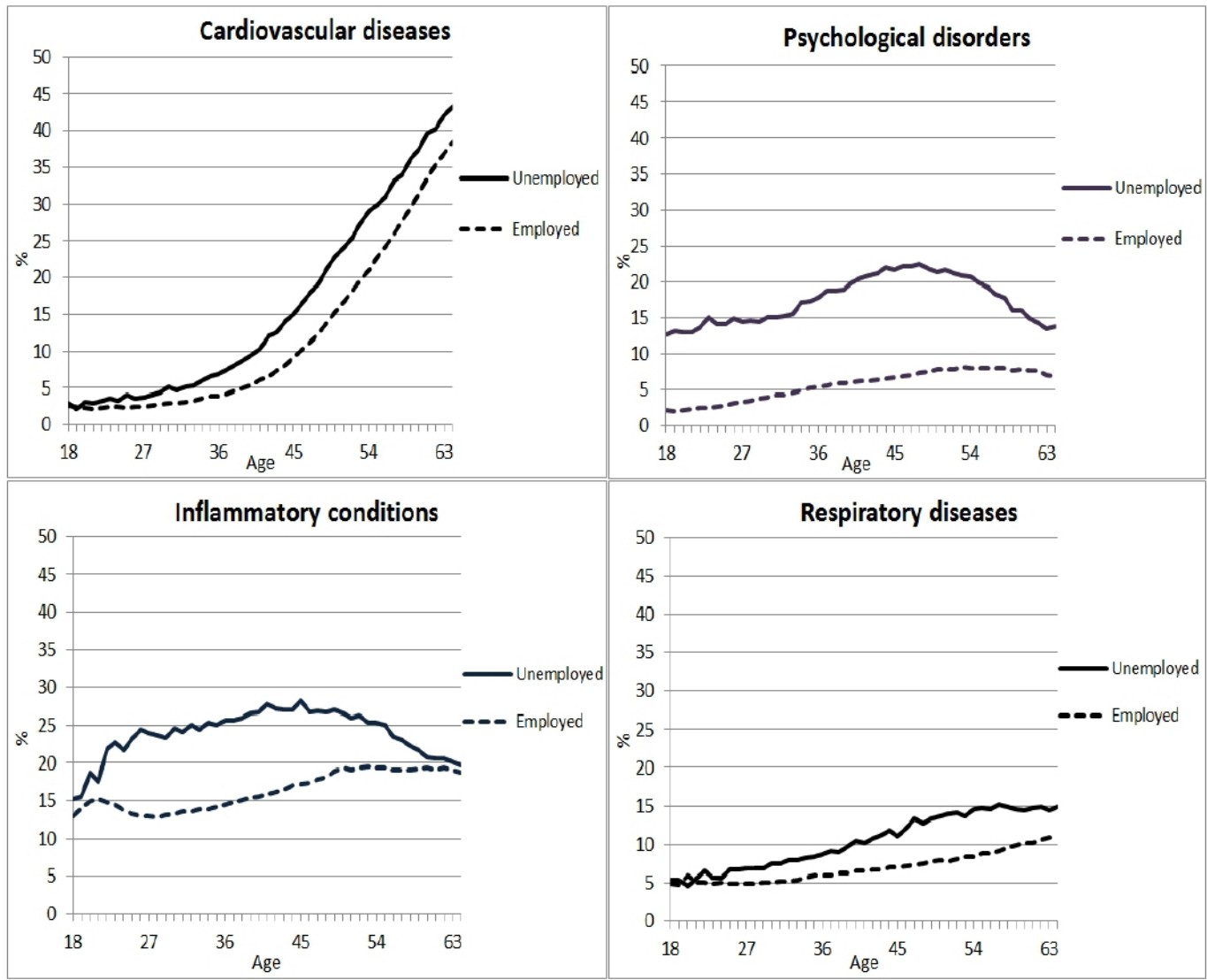

**Figure 2** Prevalence of four chronic diseases by age among unemployed (n=507 583) and employed (n=4 566 644) persons in 2016.

could also be that persons who become unemployed will deteriorate in health. Underlying mechanisms have been proposed by Jahoda who posits that unemployed persons may lack five latent functions usually observed among employed persons such as a time structure, being useful, social contacts, social status and being active.[28] The latter causation mechanism suggests that it is important that next to addressing chronic diseases, these psychosocial factors are targeted as well by interventions, in order to improve the health of unemployed persons. Improving health and employment opportunities for persons with chronic diseases is also important in the light of an ageing workforce with an expected increase of multimorbidity during the next decades.

The strength of the present study is the use of register-based data, which is a more objective method to investigate the association between health and unemployment. In earlier studies, the relationship between health and unemployment has often been examined using self-reported outcomes of health and disability.[18] However, a

major concern of self-reported health outcomes is that unemployed individuals may over-report their level of disability or work limitations to justify that they are not in paid employment.[29] In the current study, this problem has been minimised using register-based data on medication use that has been collected independently of the study. Another strength of the current study is that our register-based data capture the whole Dutch population and therefore the data provide statistical power to investigate age-specific prevalences. This facilitates precise estimations of associations between health and employment. Lastly, the use of register-based data is less expensive since no additional efforts have to be made for data collection and no concerns are present about health-related non-response.

Register-based data also have some limitations as the register only includes individuals who fulfil three criteria: (1) they are considered to need a particular drug by their general practitioner or specialist, (2) they purchase the prescribed medicine at the pharmacy and (3) the costs

**Table 2** The association of sociodemographic characteristics with the number of chronic diseases in the total population (n=5 074 227)

| | One chronic disease* | | Two chronic diseases* | | At least three chronic diseases* | |
|---|---|---|---|---|---|---|
| | n (%) | OR (95% CI) | n (%) | OR (95% CI) | n (%) | OR (95% CI) |
| **Employment status** | | | | | | |
| Employed (n=4 566 644) | 1 007 275 (22.1) | 1 | 394 396 (8.6) | 1 | 287 660 (6.3) | 1 |
| Unemployed (n=507 583) | 118 688 (23.4) | 1.30 (1.29 to 1.31) | 79 719 (15.7) | 1.74 (1.73 to 1.76) | 115 764 (22.8) | 2.59 (2.56 to 2.61) |
| **Gender** | | | | | | |
| Male (n=2 636 596) | 524 931 (19.9) | 1 | 214 039 (8.1) | 1 | 179 540 (6.8) | 1 |
| Female (n=2 437 631) | 601 032 (24.7) | 1.49 (1.49 to 1.50) | 260 076 (10.7) | 1.60 (1.59 to 1.61) | 223 884 (9.2) | 1.61 (1.60 to 1.63) |
| **Age** | | | | | | |
| 18–30 (n=1 278 018) | 249 361 (19.5) | 1 | 66 065 (5.2) | 1 | 25 443 (2.0) | 1 |
| 30–45 (n=1 965 512) | 435 651 (22.2) | 1.32 (1.31 to 1.33) | 154 922 (7.9) | 1.78 (1.77 to 1.80) | 92 888 (4.7) | 2.72 (2.68 to 2.76) |
| 45–55 (n=1 180 955) | 286 880 (24.3) | 1.80 (1.79 to 1.81) | 146 145 (12.4) | 3.38 (3.35 to 3.41) | 143 542 (12.2) | 8.00 (7.88 to 8.11) |
| 55–65 (n=649 742) | 154 071 (23.7) | 2.27 (2.26 to 2.29) | 106 983 (16.5) | 5.62 (5.55 to 5.68) | 141 551 (21.8) | 17.08 (16.83 to 17.33) |
| **Educational level** | | | | | | |
| High (n=2 096 610) | 436 035 (20.8) | 1 | 150 868 (7.2) | 1 | 89 730 (4.3) | 1 |
| Middle (n=2 077 687) | 478 089 (23.0) | 1.32 (1.31 to 1.32) | 202 938 (9.8) | 1.62 (1.60 to 1.63) | 162 416 (7.8) | 2.11 (2.09 to 2.13) |
| Low (n=899 930) | 211 839 (23.5) | 1.49 (1.48 to 1.50) | 120 309 (13.4) | 2.06 (2.04 to 2.07) | 151 278 (16.8) | 3.32 (3.29 to 3.36) |
| **Migration background** | | | | | | |
| Dutch (n=3 943 267) | 869 203 (22.0) | 1 | 356 489 (9.0) | 1 | 286 169 (7.3) | 1 |
| Moroccan (n=112 282) | 27 621 (24.6) | 1.26 (1.25 to 1.28) | 13 278 (11.8) | 1.38 (1.35 to 1.41) | 13 840 (12.3) | 1.49 (1.45 to 1.52) |
| Turkish (n=124 847) | 30 208 (24.2) | 1.34 (1.32 to 1.36) | 14 818 (11.9) | 1.58 (1.55 to 1.61) | 17 208 (13.8) | 2.03 (1.99 to 2.07) |
| Surinamese and Antillean (n=185 924) | 42 837 (23.0) | 1.10 (1.09 to 1.11) | 20 541 (11.0) | 1.19 (1.17 to 1.21) | 22 149 (11.9) | 1.39 (1.37 to 1.41) |
| Other Western (n=348 425) | 73 959 (21.2) | 0.91 (0.90 to 0.91) | 30 235 (8.7) | 0.88 (0.87 to 0.89) | 25 345 (7.3) | 0.87 (0.85 to 0.88) |
| Other Non-Western (n=359 482) | 82 135 (22.8) | 1.01 (1.00 to 1.02) | 38 754 (10.8) | 1.01 (0.99 to 1.02) | 38 713 (10.8) | 1.02 (1.01 to 1.03) |

*Persons having no chronic diseases constituted the reference group.

of the medicines are reimbursed by health insurances. For instance, persons with psychological disorders who are treated with a cognitive behavioural therapy rather than medication are not included in our analysis, and this may lead to an underestimation of persons with psychological disorders. Moreover, although a broad range of chronic diseases has been investigated in this study, several conditions that are associated with unemployment have not been included, such as back pain or musculoskeletal disorders. In the current study, it was not possible to identify these chronic conditions by the use of medication data. For instance, medication that is prescribed for back pain includes over the counter pain killers such as paracetamol or ibuprofen. However, since pain killers are used for various forms of bodily pain and no information was available regarding the reason of prescription, it was not possible to identify these health conditions. Nevertheless, it is possible that inflammatory conditions include musculoskeletal problems, as NSAIDs are a common treatment.[26] Conditions such as back pain and musculoskeletal disorders are known to lead to exit from paid employment, and therefore, should be investigated among unemployed persons in future studies.[12]

A second limitation of this study was that the cross-sectional design did not allow to gain insight into the bidirectional association between unemployment and health. However, this study provided pivotal evidence for the large differences in the prevalence of chronic diseases between unemployed and employed persons. Longitudinal or (quasi-) experimental studies are needed to further elaborate how chronic diseases lead to unemployment, and unemployment may result in chronic diseases and multimorbidity. A third limitation of this study relates to the selection of the study population of unemployed persons. Since the criteria for unemployment was defined as being unemployed for at least 9 months during a period of 1 year, our results and conclusions mainly apply to persons who are long-term unemployed. It may be that associations found in this study are less strong among short-term unemployed persons as they may have less health problems. Lastly, a limitation of the current study was the exclusion of individuals with missing data on educational level. Unemployed persons in this study more often had a lower educational level than employed person. Since there is an association between lower educational level and poorer health status, it was important to adjust for educational level in several statistical analyses. The sensitivity analysis showed comparable results in the total population and the population with educational information, indicating that education was most likely missing at random.

This study showed that health inequalities exist between unemployed and employed persons. Specifically, among the younger age group, a strong association of chronic diseases and multimorbidity with unemployment was found. Several studies have shown the beneficial effects of employment on health.[30 31] According to these studies, interventions that can support unemployed persons with chronic diseases are needed to improve employment opportunities and thus health. In order to reduce health inequalities between unemployed and employed persons, it is therefore important that reintegration policies will focus more on promoting employment among unemployed persons with chronic diseases.

In conclusion, the current study showed that unemployed persons more often have chronic diseases and multimorbidity than employed persons. The age specific prevalence follows a different pattern among employed and unemployed persons, with a relatively high prevalence of psychological disorders and inflammatory conditions among middle aged unemployed persons. Policy measures are needed to improve health and promote employment among unemployed persons.

**Contributors** BY and MS prepared the data. BY performed the statistical analysis, drafted and revised the article. MS, MGK and AB participated in the analyses. MS, MGK and AB critically reviewed the manuscript. All authors approved the final version.

**Funding** This study was funded by ZonMw (The Netherlands Organisation for Health Research and Development).

**Competing interests** None declared.

**Patient consent for publication** Not required.

**Provenance and peer review** Not commissioned; externally peer reviewed.

**Data availability statement** No data are available. This study used national registry data that can only be accessed by affiliated researchers through Remote Access.

**ORCID iDs**
Berivan Yildiz http://orcid.org/0000-0002-2385-213X
Alex Burdorf http://orcid.org/0000-0003-3129-2862

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
