## [Reviewer comments · BMJ Open]

ARTICLE DETAILS

TITLE (PROVISIONAL)	Chronic diseases and multimorbidity among unemployed and employed persons in the Netherlands: a register-based cross-sectional study
AUTHORS	Yildiz, Berivan; Schuring, Merel; Knoef, Marike; Burdorf, Alex

VERSION 1 – REVIEW

REVIEWER	Finn Breinholt Larsen Public Health and Health Service Research, DEFACTUM, Central Denmark Region Denmark
REVIEW RETURNED	16-Dec-2019

GENERAL COMMENTS	This study has two objectives: 1) To describe the prevalence of chronic diseases and multimorbidity among unemployed and employed person. 2) To investigate sociodemographic determinants of chronic diseases and multimorbidity. It is a cross-sectional study based on registry data. It concludes (page 14) that "unemployed persons more often have chronic diseases and multimorbidity than employed persons, indicating employment status to be an important determinant of health." The first part of the conclusion is hardly surprising taking into consideration the vast amount of research into ill health, sickness absence and unemployment. The second part of the conclusion is more questionable because unemployment is both associated with an increased risk of ill health and at the same time ill health can result in long-term worklessness. Due to the cross-sectional design it is not possible to sort out the relative magnitudes of these opposing effects. In the opinion of this reviewer this study only adds little to the existing knowledge on chronic disease, multimorbidity and employment status. I would like to encourage the authors to rethink the analysis in order to advance our knowledge of this important subject.
--

REVIEWER	Fiona Cocker University of Tasmania, Australia
REVIEW RETURNED	28-Jan-2020

GENERAL COMMENTS	This is a well-written paper with a rigorous study design addressing an important topic. The investigation of the prevalence of MM among unemployed persons, the first to the author's knowledge, is further strengthened by the use of register data which allows for the generalisability of the findings to the
--

	population and could effectively inform the improvement of health services for this vulnerable group. Obviously there are limitations to using cross sectional data (casual inference specifically) but this is thoroughly addressed by the authors. More discussion is required of the factors that contribute to higher prevalence of MM among unemployed persons and the identification of modifiable social and psychosocial factors could lead to health improvements. However, this is an important contribution to MM research; particularly the identification of vulnerable sub-groups within an already at risk population.
--	--

REVIEWER	Josue Almansa Department of Health Sciences University Medical Center Groningen The Netherlands
REVIEW RETURNED	22-Feb-2020

GENERAL COMMENTS	This paper describes the association between chronic diseases and unemployment with a very large sample from national registries. I would like to make some suggestions that could improve the paper.  1. The dimensionality of a statistical model is determined by the number of outcomes, and not the number of predictors. A simple regression analyses has one outcome and one predictor, and a multiple regression analyses has one outcome and several predictors. Please don't use "multivariate regression" when you mean "multiple regression". 2. Causal effects among chronic conditions and employment cannot be established in this paper. Authors already discussed this in the manuscript, but some sentences seem to be too directional and underlying a direct causality assumption. For example: "indicating employment status to be an important determinant of health." Also the word "determinant" has causal connotations. I would recommend to talk just about associations. The difficulty of interpreting causality is not only based on the cross-sectional design of the study, but also the bidirectional nature of the variables. Unemployment can cause (or initiate a reaction-chain that may lead to) some health problems, and (chronic) health problems could be a reason for unemployment. The conclusion "Policy measures and health interventions should focus more on promoting employment among unemployed persons with chronic diseases" may apply to some people, but for some chronic diseases is better to not o work (or it is just not possible). Also depends on the severity of the health condition. Specify and make clear that the suggestion about "policy measure" it's a believe from the authors, and what previous research also show, in which individuals with some chronic conditions are going to improve if they can have a suitable job - but this idea do not directly come from the results of this study. 3. In limitations it's somehow mentioned that there is no clinical diagnostic, but only an inference based on the registered medication. Also, with this method, the severity of the health condition cannot be (easily) assessed. 4. The large sample size it's indeed and advantage. Authors mentioned "providing enormous statistical power and thus smaller confidence intervals", but there is no p-value or confidence interval
---

	showed in this paper. And I even think that it's a disadvantage for the classical statistical tests, because almost every test would be significant even in the absence of relevant clinical differences. So I would delete that. I think the big advantage of having such large sample offers the possibility of estimations across very specific subgroups, for example, the age specific prevalences and associations. 5a. It was not clear why education variable was not measured for large part of the sample, and why this variable is so important that individuals without it are going to be excluded. 5b. I'm still not sure what to suggest about how to use the education variable. My first impression was to use always as much data as possible, and only loose the individuals with missing education when using this variable in a statistical model. Authors decided to use always the sample with education, which also make sense to me, because results are more comparable given that uses always the same sample - and as a sensitivity analyses it is mentioned in the discussion that adding the individuals without education would lead to same results, thus the assumption of missing at random make sense. Both approaches make sense to me, but I would suggest to make clear the reasoning for their choice. 6. In the methods section several times says that some results "were presented" (for example: "Chronic diseases with a prevalence higher than 5% were presented."). Not sure if this means that the tables will only show part of the results (in that case you don't have to mention this in the methods section) or if some individuals where excluded for some estimations. 6b. Why only estimate the medicines specific per age for cardio. and psych. disorders (and not the others chronic conditions)? 7a. Statistical methods: Seems that you choose no-disorder as reference category. If so, make it explicit. 7b. "The association between employment status and multimorbidity stratified by age and educational level was also investigated." I see in supplementary table 3 the stratification by age groups, but I think there is no results stratifying by education. 8. Tables should be the more self-explanatory as possible. It's not clear across all tables and figures what the percentages refer to (percentage respect to what?) In Table 1 I miss the results for no chronic conditions. Table 2. is this simple or multiple regression? Also I assume that the reference category for number of chronic conditions is zero (no-disorder group). Btw, I like figure 1. 9a. Results: In second line, the 48.1% appears twice. I would say this is a mistake, maybe the "compared to employed persons" should be 19.4%? 9b. In the paragraph starting by "At all ages," respiratory conditions seems to increase with age more then inflammatory (figure 2).. 10. When talking about co-occurrence (or even about results for specific chronic conditions), it's not always clear if authors refer to them as "exclusive" disorders or if other disorders may be
--	--

	included. Mostly seems like results show disorders including all other possible comorbidities. Make it clear in the text is this is what you mean. For example, "The co-occurrence of both psychological disorders and inflammatory conditions was higher among unemployed persons (6.2%) than among employed persons (1.5%)." does not refer to only both psychological disorders and inflammatory, but also when this 2 disorders my appear together with other ones. 10b. Discussion: "Between unemployed and employed persons, the largest differences were observed for cardiovascular diseases and psychological disorders". Looking at figure 2 seems no much difference in cardio. Maybe you refer to supplementary table 2 with "exclusive" only cardio and only psycho? (while figure 2 is maybe non-exclusive?). 11: Which results supports this conclusion: "...also among other groups, such as disabled individuals, a decline was observed." Where can we see the results for 'disabled'? and what do you mean by disabled? Also not clear what do you mean by 'selection' in "...selection of unemployed individuals could explain the decline...". Another possible explanation is that individuals with health conditions have lower life expectancy, and this is why the prevalences at older ages decreases (only the healthiest remain).
--	--

REVIEWER	Mònica Ubalde López Barcelona Institute for Global Health, Spain
REVIEW RETURNED	27-Feb-2020

GENERAL COMMENTS	COMMENTS TO THE AUTHOR: The author presents a study that investigates prevalence an determinants of multimorbidity in working-age general Dutch population in two employment stages (employed and unemployed), which is of high relevance as related evidence to the workforce is still scarce. However, there are several flaws in the methods and description of results that need to be addressed. Also, a more developed and deeper discussion section is required. After reading carefully the manuscript I had to conclude that it cannot be published in its current form and I recommend a major revision. I would like to provide some general comments and suggestions to the authors that I hope to be helpful and useful to improve the paper. C1-Based on the definition of multimorbidity (i.e., the co-existence of two or more chronic conditions) I suggest to delete "chronic conditions" from the title . C2- I suggest to reformulate the second objective in the abstract and the introduction section as : "to examine associations between employment status and sociodemographic characteristics with chronic diseases and multimorbidity" C3- Why are 9 months the criteria to classified participants into the employed or unemployed group? What are the criteria behind this
---

	time period? My concern is that people in unemployment for more than 9 months could be considered as long-term unemployed and more likely to have worse health status (more and more severe chronic conditions) than employed, even than people with periods of unemployment below 9 months. Also, information of medication use relates to the more severe health-related conditions. This leads to a selection bias that needs to be considered when interpreting the results. C4- The definition of multimorbidity is unclear as well as the criteria for selection of chronic conditions included. On one hand, under the title "Chronic diseases and multimorbidity" (page 5, methods) the author refers to identify 21 chronic conditions from the ATC-codes that are grouped as Multimorbidity from none to ≥ 3 chronic diseases. Having no chronic condition or 1 cannot be called multimorbidity. On the second hand, under the title "Analyses" (page 6, methods) the author describes multimorbidity as: "...the potential proportion of individuals with all potential combinations of four chronic diseases...". Why four diseases out of the previous 21 identified, and why the four selected? I don't feel the study of the prevalence of combinations with 4 chronic conditions as study of multimorbidity prevalence, but as kind of study of 4 chronic conditions clustering. C5- Based on that nature of the categories I think it would be more correct and accurate to use term Nationality rather than Ethnicity (i.e. Nationality: refers to the country that a person belongs to either by birth right or naturalization. Ethnicity: is a category of people who share a heritage based on race, language, or culture) C6-Missing data excluded (33% of eligible population) is not mentioned in the methods section C7- The numbers for the prevalence described in the top paragraph on page 8 do not fit with those in Figure 1 nor Supplementary Table 2 C8- The study has critical limitations that need to be discussed deeper to better interpret the results: 1) the reverse causality (cross-sectional design) together with the healthy worker effect (selection bias); 2) classification of employment status based on 9 months (potential misclassification bias). It is likely that those classified into unemployment in 2016 have had a prior low labor market participation because of their poor health status (more severe health-conditions); 3) Together with common mental disorders, musculoskeletal diseases are not included even though being the most prevalent chronic conditions that account for multimorbidity prevalence. However, it is possible that inflammatory conditions include musculoskeletal problems as NSAID's are a common treatment. In this sense the potential explanation for the decrease in prevalence of inflammatory conditions among older ages could be underestimated. C9-All tables and figures should be at the end of the main text. C10-Table 1. I suggest to add p-values for differences between the two groups prevalence.
--	--

	C11-Table 2. I suggest to place the columns with prevalence for 1,2 and >= chronic conditions in Table 1 (descriptive). C12-Supplementary Table 2: The first 4 rows are single chronic diseases but no multimorbidity as the title of the table states. C13- No ethical issues are addressed
--	--

VERSION 1 – AUTHOR RESPONSE

Reviewers' Comments to Author:

Reviewer: 1

It is a cross-sectional study based on registry data. It concludes (page 14) that "unemployed persons more often have chronic diseases and multimorbidity than employed persons, indicating employment status to be an important determinant of health." The first part of the conclusion is hardly surprising taking into consideration the vast amount of research into ill health, sickness absence and unemployment. The second part of the conclusion is more questionable because unemployment is both associated with an increased risk of ill health and at the same time ill health can result in long-term worklessness. Due to the cross-sectional design it is not possible to sort out the relative magnitudes of these opposing effects.

In the opinion of this reviewer this study only adds little to the existing knowledge on chronic disease, multimorbidity and employment status. I would like to encourage the authors to rethink the analysis in order to advance our knowledge of this important subject.

We thank the reviewer for the critical comments on our paper. Below our answers.

Response to the comment:

We acknowledge that there has been a number of studies showing that unemployed persons have poorer health than employed persons. However, it is noteworthy to mention that our study contributes to the existing knowledge in several ways. First, most studies have used self-reports on health, as we have reported in a systematic review a few years ago. (1) In recent years, only a few studies have used more objective information, which is important, for example because of justification bias. Second, only very few studies provided insight into a broader range of health aspects. Many of existing studies mainly focused on certain aspects of health such as mental health (e.g depression). One of the major strengths of our study is that we are able to investigate a broad range of chronic diseases such as diabetes, inflammatory conditions and respiratory illness. Third, our study contributes to the scarce amount of literature on the prevalence of multimorbidity among unemployed persons, using objective medication data rather than subjective self-reported health outcomes. Lastly, our study is further strengthened by the use of large register data which allows for the generalizability of the findings to the population and which offers us the possibility to investigate specific subgroups (e.g. age specific prevalences and associations).

These arguments are presented in the introduction of the paper.

Since we use a cross-sectional design, it falls outside the scope of our study to investigate or conclude whether ill health results in unemployment or unemployment causes ill health. However, we do add to the literature by comparing age specific prevalence of several chronic diseases between unemployed and employed persons. We therefore changed the second part of the conclusion in the abstract and the discussion:

Old text (Abstract, p. 1, line 25):

Conclusion: Large inequalities exist in the prevalence of chronic diseases and multimorbidity among unemployed and employed persons, indicating employment status to be an important determinant of health.

New text:

Large differences exist in the prevalence of chronic diseases and multimorbidity between unemployed and employed persons. The age specific prevalence follows a different pattern among employed and

unemployed persons, with a relatively high prevalence of psychological disorders and inflammatory conditions among middle aged unemployed persons.

Old text (Discussion, p. 16, line 1):

In conclusion, the current study showed that unemployed persons more often have chronic diseases and multimorbidity than employed persons, indicating employment status to be an important determinant of health.

New Text:

In conclusion, the current study showed that unemployed persons more often have chronic diseases and multimorbidity than employed persons. The age specific prevalence follows a different pattern among employed and unemployed persons, with a relatively high prevalence of psychological disorders and inflammatory conditions among middle aged unemployed persons.

Reviewer: 2

Comments to the author

This is a well-written paper with a rigorous study design addressing an important topic. The investigation of the prevalence of MM among unemployed persons, the first to the author's knowledge, is further strengthened by the use of register data which allows for the generalisability of the findings to the population and could effectively inform the improvement of health services for this vulnerable group. Obviously there are limitations to using cross sectional data (casual inference specifically) but this is thoroughly addressed by the authors. More discussion is required of the factors that contribute to higher prevalence of MM among unemployed persons and the identification of modifiable social and psychosocial factors could lead to health improvements. However, this is an important contribution to MM research; particularly the identification of vulnerable sub-groups within an already at risk population.

Response to the comment

We thank the reviewer for the positive comments and for acknowledging the relevance of this study for research and policy. We included a paragraph in the discussion about the factors that may contribute to the higher prevalence of multimorbidity among unemployed persons, and how these factors could lead to health improvements.

New text (Discussion, p.13, line 29):

Unemployed persons had a higher prevalence of multimorbidity than employed persons. It is likely that the healthy worker selection process is more prominent among persons with multiple diseases than single diseases. (27) According to the causation mechanism, it could also be that persons who become unemployed will deteriorate in health. Underlying mechanisms have been proposed by Jahoda who posits that unemployed persons may lack five latent functions usually observed among employed persons such as a time structure, being useful, social contacts, social status and being active. (28) The latter causation mechanism suggests that it is important that next to addressing chronic diseases, these psychosocial factors are targeted as well by interventions, in order to improve the health of unemployed persons. Improving health and employment opportunities for persons with chronic diseases is also important in the light of an aging workforce with an expected increase of multimorbidity during the next decades.

Reviewer: 3

Comment to the author

This paper describes the association between chronic diseases and unemployment with a very large sample from national registries. I would like to make some suggestions that could improve the paper.

Comment 1

The dimensionality of a statistical model is determined by the number of outcomes, and not the number of predictors. A simple regression analyses has one outcome and one predictor, and a multiple regression analyses has one outcome and several predictors. Please don't use "multivariate regression" when you mean "multiple regression".

Response to comment 1

We have replaced multivariate by multiple regression analysis throughout the paper.

Comment 2

Causal effects among chronic conditions and employment cannot be established in this paper. Authors already discussed this in the manuscript, but some sentences seem to be too directional and underlying a direct causality assumption. For example: "indicating employment status to be an important determinant of health." Also the word "determinant" has causal connotations. I would recommend to talk just about associations. The difficulty of interpreting causality is not only based on the cross-sectional design of the study, but also the bidirectional nature of the variables. Unemployment can cause (or initiate a reaction-chain that may lead to) some health problems, and (chronic) health problems could be a reason for unemployment. The conclusion "Policy measures and health interventions should focus more on promoting employment among unemployed persons with chronic diseases" may apply to some people, but for some chronic diseases is better to not o work (or it is just not possible). Also depends on the severity of the health condition. Specify and make clear that the suggestion about "policy measure" it's a believe from the authors, and what previous research also show, in which individuals with some chronic conditions are going to improve if they can have a suitable job - but this idea do not directly come from the results of this study.

Response to comment 2

We appreciate the comment and agree with the reviewer that due to the cross-sectional design and the bidirectional nature of health and employment it is not possible to explore causal relations. We have rephrased the following sentences:

Old text (Abstract, p. 1, line 25):

Conclusion: Large inequalities exist in the prevalence of chronic diseases and multimorbidity among unemployed and employed persons, indicating employment status to be an important determinant of health.

New text:

Conclusion: Large differences exist in the prevalence of chronic diseases and multimorbidity among unemployed and employed persons. The age specific prevalence follows a different pattern among employed and unemployed persons, with a relatively high prevalence of psychological disorders and inflammatory conditions among middle aged unemployed persons.

Old text (Discussion, p. 16, line 1):

In conclusion, the current study showed that unemployed persons more often have chronic diseases and multimorbidity than employed persons, indicating employment status to be an important determinant of health.

New Text:

In conclusion, the current study showed that unemployed persons more often have chronic diseases and multimorbidity than employed persons. The age specific prevalence follows a different pattern among employed and unemployed persons, with a relatively high prevalence of psychological disorders and inflammatory conditions among middle aged unemployed persons.

We have now included the cross-sectional design as a limitation rather than shortly mentioning it at the end of the discussion:

New text (Discussion, p. 15, line 7):

A second limitation of this study was that the cross-sectional design did not allow to gain insight into the bi-directional association between unemployment and health. However, this study provided pivotal evidence for the large differences in the prevalence of chronic diseases between unemployed and employed persons. Longitudinal or (quasi-) experimental studies are needed to further elaborate how chronic diseases lead to unemployment, and unemployment may result in chronic diseases and multimorbidity.

Old text (Discussion, p. 15, line 24):

This study showed that health inequalities exist between unemployed and employed persons. Specifically, among the younger age group, a strong association of chronic diseases and multimorbidity with unemployment was found. Due to the cross-sectional design, it was not possible to investigate causal relationships between unemployment and health. However, several studies have shown beneficial effects of employment on health. Although it may be a challenge to increase employment rates among unemployed persons with chronic diseases, it may lead to improvements in health. In order to reduce health inequalities between unemployed and employed persons, it is therefore important that re-integration policies focus more on promoting employment among unemployed persons with chronic diseases.

New text:

This study showed that health inequalities exist between unemployed and employed persons. Specifically, among the younger age group, a strong association of chronic diseases and multimorbidity with unemployment was found. Several studies have shown the beneficial effects of employment on health. (30,31) *According to these studies, interventions that can support unemployed persons with chronic diseases are needed to improve employment opportunities and thus health.* In order to reduce health inequalities between unemployed and employed persons, it is therefore important that re-integration policies *will* focus more on promoting employment among unemployed persons with chronic diseases.

Comment 3

In limitations it's somehow mentioned that there is no clinical diagnostic, but only an inference based on the registered medication. Also, with this method, the severity of the health condition cannot be (easily) assessed.

Response to comment 3

Indeed, a limitation is that we were not able to assess the severity of the health conditions. However, there is certainly an underlying clinical diagnosis as medication was prescribed by a physician. We do not include over the counter medication. We acknowledge that it is of course possible to suffer from a medically diagnosed disease without being prescribed a drug. We have added this to the discussion:

New text (Discussion, p. 14, line 22):

Register-based data also have some limitations as the register only includes individuals who fulfill three criteria: 1) they are considered to need a particular drug by their general practitioner or specialist, 2) they purchase the prescribed medicine at the pharmacy, and 3) the costs of the medicines are reimbursed by health insurances. For instance, persons with psychological disorders who are treated with a cognitive behavioral therapy rather than medication are not included in our analysis, and this may lead to an underestimation of persons with psychological disorders.

Comment 4

The large sample size it's indeed an advantage. Authors mentioned "providing enormous statistical power and thus smaller confidence intervals", but there is no p-value or confidence interval showed in this paper. And I even think that it's a disadvantage for the classical statistical tests, because almost every test would be significant even in the absence of relevant clinical differences. So I would delete that. I think the big advantage of having such a large sample offers the possibility of estimations across very specific subgroups, for example, the age-specific prevalences and associations.

Response to comment 4

We thank the reviewer for this critical methodological comment. We clarified that the large sample size allowed us to investigate age-specific prevalences:

Old text (Discussion, p. 14, line 15):

Another strength of the current study is that in our register-based data the whole Dutch population is involved and therefore the data provide enormous statistical power resulting in smaller confidence intervals.

New text (Discussion, p. 14, line 15):

Another strength of the current study is that our register-based data capture the whole Dutch population and therefore the data provide statistical power to investigate age-specific prevalences.

Comment 5a

It was not clear why education variable was not measured for a large part of the sample, and why this variable is so important.

Response to comment 5a

We thank the reviewer for this comment. We clarified in the Methods why this was not measured for a large part of the sample:

New text (Methods, p. 5, line 11):

Individuals aged between 18 and 64 years with available information on employment status were selected (n=10,514,271). This selection captured individuals who were not eligible for exit from paid employment through statutory national retirement schemes. Due to the lack of nationwide education registers in the past, many older persons had missing data on educational level. Also, the current register only includes formal education obtained at institutes financed by government. Therefore, we excluded 31.9% of individuals with missing data on educational level (n=3,356,002).

Comment 5b

I'm still not sure what to suggest about how to use the education variable. My first impression was to use always as much data as possible, and only lose the individuals with missing education when using this variable in a statistical model. Authors decided to use always the sample with education, which also makes sense to me, because results are more comparable given that we always use the same sample - and as a sensitivity analysis it is mentioned in the discussion that adding the individuals without education would lead to the same results, thus the assumption of missing at random makes sense. Both approaches make sense to me, but I would suggest to make clear the reasoning for their choice.

Response to comment 5b

Thank you for sharing your thoughts on both approaches. We shortly clarified in the discussion why we chose to exclude individuals with missing data on educational level:

New text (Discussion, p.15, line 16):

Lastly, a limitation of the current study was the exclusion of individuals with missing data on educational level. Unemployed persons in this study more often had a lower educational level than employed persons. Since there is an association between lower educational level and poorer health status, it was important to adjust for educational level in several statistical analyses. The sensitivity analysis showed comparable results in the total population and the population with educational information, indicating that education was most likely missing at random.

Comment 6

In the methods section several times says that some results "were presented" (for example: "Chronic diseases with a prevalence higher than 5% were presented."). Not sure if this means that the tables will only show part of the results (in that case you don't have to mention this in the methods section) or if some individuals were excluded for some estimations.

Response to comment 6

We agree with the reviewer that it is not necessary to mention this sentence, since our results indeed show part of the results. We therefore removed the following sentence from the methods section:

Chronic diseases with a prevalence higher than 5% were presented.

Comment 6b

Why only estimate the medicines specific per age for cardio. and psych. disorders (and not the others chronic conditions)?

Response to comment 6b

Since we had data on specific medicines for each chronic disease, we found it useful to distinguish between specific conditions within a chronic disease. Specifying several conditions within a chronic diseases could be done more accurately for CVD and psychological disorders, (as presented in Supplementary Table 1). This is to a lesser extent the case for the other chronic diseases, since less specific medicines can be distinguished. We shortly clarified this in the Methods:

Old text (Methods, p. 7, line 10):

(..) for cardiovascular diseases and psychological disorders, the age-specific prevalence of specific medicines used was also presented.

New text (Methods, p. 7, line 10):

In order to distinguish specific conditions within a chronic disease, the age-specific prevalence of specific medicines was also investigated for cardiovascular diseases and psychological disorders. The latter was not investigated for the other chronic diseases because less specific medicines could be distinguished within a chronic disease.

Comment 7a

Statistical methods: Seems that you choose no-disorder as reference category. If so, make it explicit.

Response to comment 7a

In the methods, we added the following sentence:

In these analyses, having no chronic diseases was used as the reference category.

Comment 7b

"The association between employment status and multimorbidity stratified by age and educational level was also investigated."

I see in supplementary table 3 the stratification by age groups, but I think there is no results stratifying by education.

Response to comment 7b

Thank you for this remark. We indeed only have results stratified by age, and therefore deleted 'educational level' from this sentence.

Comment 8

Tables should be the more self-explanatory as possible.

It's not clear across all tables and figures what the percentages refer to (percentage respect to what?) In Table 1 I miss the results for no chronic conditions.

Response to comment 8

We have adjusted the layout of Table 2, and added the results of the number of chronic diseases to Table 1.

Addition to Table 1 (Results, p. 11):

Number of chronic diseases	Unemployed (n=507,583)	Employed (n=4,566,644)
	n (%)	n (%)
0	193,412 (38.1)	2,877,313 (63.0)
1	118,688 (23.4)	1,007,275 (22.1)
2	79,719 (15.7)	394,396 (8.6)
≥3	115,764 (22.8)	287,660 (6.3)

Comment Table 2

is this simple or multiple regression? Also I assume that the reference category for number of chronic conditions is zero (no-disorder group).

Btw, I like figure 1.

Response to comment on Table 2

We thank the reviewer for the positive comment on figure 1. We clarified in the results as a footnote in Table 2 that no-disorder is used as the reference group:

*persons having no chronic diseases constituted the reference group.

In the Methods, we added that the analyses were adjusted for several factors:

New Text (Methods, p. 7, line 1)

Logistic regression analyses were adjusted for age, sex, educational level and migration background.

Comment 9a

Results: In second line, the 48.1% appears twice. I would say this is a mistake, maybe the "compared to employed persons" should be 19.4%?

Response to comment 9a

We appreciate the attentive comment, and replaced it by 19.4%.

Comment 9b

In the paragraph starting by "At all ages," respiratory conditions seems to increase with age more than inflammatory (figure 2)..

Response to comment 9b

Thank you for this attentive comment, we have rephrased the sentence:

New text (Result, p. 9, line 15):

The prevalence of cardiovascular diseases and *respiratory diseases* increased with age among both unemployed and employed persons.

Comment 10

When talking about co-occurrence (or even about results for specific chronic conditions), it's not always clear if authors refer to them as "exclusive" disorders or if other disorders may be included. Mostly seems like results show disorders including all other possible comorbidities. Make it clear in the text is this is what you mean.

For example, "The co-occurrence of both psychological disorders and inflammatory conditions was higher among unemployed persons (6.2%) than among employed persons (1.5%)." does not refer to only both psychological disorders and inflammatory, but also when this 2 disorders my appear together with other ones.

Response to comment 10

We agree that it may not always be clear if other disorders are included or not. We clarified this in the Methods:

Old Text (Methods, p. 6, line 6):

The presence of a chronic disease was dichotomized into having or not having a chronic condition. The total number of chronic diseases was computed for each participant. This measure of multimorbidity of chronic diseases was categorized into four groups: no chronic disease, one chronic disease, two chronic diseases, and at least three chronic diseases. Chronic diseases with a prevalence higher than 5% were presented. The total number of chronic diseases included also those with a prevalence lower than 5%, capturing all 21 chronic diseases. All 21 chronic diseases with identifying medications are described by Huber et al.

New text (Methods, p. 6, line 6):

The presence of a specific chronic disease was dichotomized into having or not having a chronic disease. Multimorbidity was investigated as 1) the number of chronic diseases and 2) the combinations of four common chronic diseases with the highest prevalence in the study population. For the first approach of multimorbidity, the total number of chronic diseases was computed for each participant, based on 21 different chronic diseases that could be identified by medication prescription. (13) This measure of multimorbidity was categorized into four groups: no chronic disease, one chronic disease, two chronic diseases, and at least three chronic diseases. For the second approach, we used the following four chronic diseases to describe their co-occurrence: cardiovascular diseases, psychological disorders, inflammatory conditions, and respiratory diseases.

Comment 10b

Discussion: "Between unemployed and employed persons, the largest differences were observed for cardiovascular diseases and psychological disorders". Looking at figure 2 seems no much difference in cardio. Maybe you refer to supplementary table 2 with "exclusive" only cardio and only psycho? (while figure 2 is maybe non-exclusive?).

Response to comment

Thank you for the critical comment. We explained in the discussion why in Figure 2 the difference seems not to be so large:

Old Text (Discussion, p. 12, line 14):

The higher prevalence of chronic diseases among unemployed persons in the current study is in line with other studies showing a negative association between unemployment and health. Several studies have shown that unemployed persons have a poorer mental and physical health status. For instance, unemployed persons had high risks of common mental disorders such as depression. Our study added to the current literature by comparing the age-specific prevalence of chronic diseases between unemployed and employed persons.

New text (Discussion, p. 12, line 14):

The higher prevalence of chronic diseases among unemployed persons in the current study is in line with other studies showing that unemployed persons have a poorer mental and physical health status. (2,5,18) For instance, unemployed persons had high risks of common mental disorders such as depression. (19-21) Our study added to the current literature by comparing the age-specific prevalence of chronic diseases between unemployed and employed persons. *Between unemployed and employed persons, the largest differences in prevalence were observed for cardiovascular diseases and psychological disorders. Although the overall prevalence of cardiovascular diseases was much higher among unemployed persons, the age-specific patterns showed small differences, indicating that the higher age among unemployed persons was largely responsible for the higher prevalence of cardiovascular diseases.*

Comment 11

Which results supports this conclusion: "...also among other groups, such as disabled individuals, a decline was observed." Where can we see the results for 'disabled'? and what do you mean by disabled? Also not clear what do you mean by 'selection' in "...selection of unemployed individuals could explain the decline..."

Another possible explanation is that individuals with health conditions have lower life expectancy, and this is why the prevalences at older ages decreases (only the healthiest remain).

Response to comment 11

With 'disabled' we mean individuals who receive a disability benefit. We additionally performed the analyses of figure 2 among individuals who received a disability benefit for at least 9 months in 2016. By this, we aimed to investigate whether the decrease in psychological disorders from middle age onwards could be due to individuals being unemployed. These results are not shown, and this is clarified in the discussion. With 'selection', we mean our study population of unemployed persons. We understand that this may lead to confusion.

Old text (Discussion, p. 13, line 8):

(..) It was checked whether the selection of unemployed individuals could explain the decline in the prevalence of antidepressants use, anxiolytics, hypnotics and sedatives after the age of 50. This does not appear to be the case as also among other groups, such as disabled individuals, a decline was observed.

New text (Discussion, p. 13, line 8):

It was checked whether the decline in the prevalence of use of antidepressants, anxiolytics, hypnotics and sedatives after the age of 50 would be different among persons receiving a disability benefit. Namely, it might be possible that older unemployed persons with chronic diseases are more likely to receive disability benefits rather than unemployment or social benefits, and therefore the age-specific prevalence of these medicines may decline among unemployed persons. However, also among persons with disability benefits, a decline was observed from middle age onwards for these medicines (results not shown).

Reviewer: 4

Comment author

The author presents a study that investigates prevalence and determinants of multimorbidity in working-age general Dutch population in two employment stages (employed and unemployed), which is of high relevance as related evidence to the workforce is still scarce.

However, there are several flaws in the methods and description of results that need to be addressed. Also, a more developed and deeper discussion section is required.

After reading carefully the manuscript I had to conclude that it cannot be published in its current form and I recommend a major revision.

I would like to provide some general comments and suggestions to the authors that I hope to be helpful and useful to improve the paper.

Response to the comment

We thank the reviewer for the critical comments that helped us to improve the paper. See below our answers.

Comment 1

Based on the definition of multimorbidity (i.e., the co-existence of two or more chronic conditions) I suggest to delete "chronic conditions" from the title.

Response to comment 1

We agree with the reviewer, and changed the title to:

Chronic diseases and multimorbidity among unemployed and employed persons in the Netherlands: a register-based cross-sectional study

Comment 2

I suggest to reformulate the second objective in the abstract and the introduction section as : "to examine associations between employment status and sociodemographic characteristics with chronic diseases and multimorbidity"

Response to comment 2

We rephrased this sentence also in the abstract:

Old text (Abstract, p. 1, line 4):

The second objective was to investigate sociodemographic determinants of chronic diseases and multimorbidity.

New text (Abstract, p.1, line 4):

The second objective was to examine associations of employment status and sociodemographic characteristics with chronic diseases and multimorbidity.

Comment 3

Why are 9 months the criteria to classified participants into the employed or unemployed group? What are the criteria behind this time period? My concern is that people in unemployment for more than 9 months could be consider as in long-term unemployed and more likely to have worst health status (more and more severe chronic conditions) than employed, even than people with periods of unemployment below 9 moths. Also, information of medication use relates to the more severe health-related conditions. This leads to a selection bias that needs to be consider when interpreting the results.

Response to comment 3

We acknowledge that our results merely apply to long-term unemployed. This is a limitation of our study and is addressed in the discussion:

New text (Discussion, p. 15, line 12):

A third limitation of this study relates to the selection of the study population of unemployed persons. Since the criteria for unemployment was defined as being unemployed for at least 9 months during a period of one year, our results and conclusions mainly apply to persons who are long-term unemployed. It may be that associations found in this study are less strong among short-term unemployed persons as they may have less health problems.

Comment 4

The definition of multimorbidity is unclear as well as the criteria for selection of chronic conditions included. On one hand, under the title "Chronic diseases and multimorbidity"(page 5, methods) the author refers to identify 21 chronic conditions from the ATC-codes that are grouped as Multimorbidity from none to ≥ 3 chronic diseases. Having no chronic condition or 1 cannot be called multimorbidity. On the second hand, under the title "Analyses" (page 6, methods) the author describes multimorbidity as: "...the potential proportion of individuals with all potential combinations of four chronic diseases...". Why four diseases out of the previous 21 identified, and why the four selected? I don't feel the study of the prevalence of combinations with 4 chronic conditions as study of multimorbidity prevalence, but as kind of study of 4 chronic conditions clustering.

Response to comment 4

We thank the reviewer for addressing this comment on the definition of multimorbidity. We have explained why and how we operationalized multimorbidity in two ways:

Old Text (Methods, p. 6, line 6):

The presence of a chronic disease was dichotomized into having or not having a chronic condition. The total number of chronic diseases was computed for each participant. This measure of multimorbidity of chronic diseases was categorized into four groups: no chronic disease, one chronic disease, two chronic diseases, and at least three chronic diseases. Chronic diseases with a prevalence higher than 5% were presented. The total number of chronic diseases included also those with a prevalence lower than 5%, capturing all 21 chronic diseases. All 21 chronic diseases with identifying medications are described by Huber et al.

New text (Methods, p. 6, line 6):

The presence of a specific chronic disease was dichotomized into having or not having a chronic disease. Multimorbidity was investigated as 1) the number of chronic diseases and 2) the combinations of four common chronic diseases with the highest prevalence in the study population. For the first approach of multimorbidity, the total number of chronic diseases was computed for each participant, based on 21 different chronic diseases that could be identified by medication prescription. (13) This measure of multimorbidity was categorized into four groups: no chronic disease, one chronic disease, two chronic diseases, and at least three chronic diseases. For the second approach, we used the following four chronic diseases to describe their co-occurrence: cardiovascular diseases, psychological disorders, inflammatory conditions, and respiratory diseases.

Comment 5

Based on that nature of the categories I think it would be more correct and accurate to use term Nationality rather than Ethnicity (i.e. Nationality: refers to the country that a person belongs to either by birth right or naturalization. Ethnicity: is a category of people who share a heritage based on race, language, or culture)

Response to comment 5

We cannot use the term nationality since the data does not provide us information on the nationality of persons but only on the country of birth. From this information we cannot conclude whether one person has the nationality of a particular country. We think that migration background better fits the data rather than nationality or ethnicity and therefore replaced ethnicity by migration background throughout the paper.

Comment 6

Missing data excluded (33% of eligible population) in not mentioned in the methods section

Response to comment 6 (nog toevoegen)

We have indeed not mentioned the percentage of missings in the methods section. The number of missings is now added to the Methods:

New text (Methods, p. 5, line 11):

Individuals aged between 18 and 64 years with available information on employment status were selected (n=10,514,271). This selection captured individuals who were not eligible for exit from paid employment through statutory national retirement schemes. Due to the lack of nationwide education registers in the past, many older persons had missing data on educational level. Also, the current register only includes formal education obtained at institutes financed by government. Therefore, we excluded 31.9% of individuals with missing data on educational level (n=3,356,002). Within the population with available data on all sociodemographic characteristics (n=7,158,269), 4,566,644 persons were classified as employed and 507,583 persons were classified as unemployed. In total, 5,074,227 subjects were included in the present study.

Comment 7

The numbers for the prevalence described in the top paragraph on page 8 do not fit with those in Figure 1 nor Supplementary Table 2

Response to comment 7

We have added the sum of the co-occurrence of psychological disorders and inflammatory conditions in the text of the paper to give the reader some directions:

Addition (Results, p. 9, line 4):

The prevalence of multimorbidity was higher for unemployed persons compared to employed persons. The co-occurrence of both psychological disorders and inflammatory conditions was higher among unemployed persons ($3.6\%+0.9\%+1.2\%+0.5\%=6.2\%$) than among employed persons ($1.0\%+0.2\%+0.2\%+0.1\%=1.5\%$). In addition, the co-occurrence of cardiovascular diseases and inflammatory conditions was higher among unemployed persons (5.5%) compared to employed persons (2.1%). The prevalence of having both cardiovascular diseases and psychological disorders was 4.9% among unemployed persons compared to 0.9% among employed persons. (Figure 1, Supplementary Table 2)

Comment 8

The study has critical limitations that need to be discussed deeper to better interpret the results: 1) the reverse causality (cross-sectional design) together with the healthy worker effect (selection bias); 2) classification of employment status based on 9 months (potential misclassification bias). It is likely that those classified into unemployment in 2016 have had a prior low labor market participation because of their poor health status (more severe health-conditions); 3) Together with common mental disorders, musculoskeletal diseases are not included even though being the most prevalent chronic conditions that account for multimorbidity prevalence. However, it is possible that inflammatory conditions include musculoskeletal problems as NSAID's are a common treatment. In this sense the

potential explanation for the decrease in prevalence of inflammatory conditions among older ages could be underestimated.

Response to comment 8

Thank you for these critical comments, which helped to improve the discussion of the paper. We have incorporated the first comment in a paragraph in the discussion:

New text (Discussion, p. 13, line 29):

Unemployed persons had a higher prevalence of multimorbidity than employed persons. It is likely that the healthy worker selection process is more prominent among persons with multiple diseases than single diseases. (27) According to the causation mechanism, it could also be that persons who become unemployed will deteriorate in health. Underlying mechanisms have been proposed by Jahoda who posits that unemployed persons may lack five latent functions usually observed among employed persons such as a time structure, being useful, social contacts, social status and being active. (28) The latter causation mechanism suggests that it is important that next to addressing chronic diseases, these psychosocial factors are targeted as well by interventions, in order to improve the health of unemployed persons. Improving health and employment opportunities for persons with chronic diseases is also important in the light of an aging workforce with an expected increase of multimorbidity during the next decades.

We have addressed the second comment of the author in a previous response (see above comment 3).

Regarding the third comment, we would like to mention that we identified psychological disorders based on three types of medication: antidepressants, anxiolytics, and hypnotics and sedatives. Psychotic disorders were identified based on antipsychotics used in schizophrenia and bipolar disorders. In this way, we were able to investigate different types of common mental disorders. This has already been addressed in the paper.

Furthermore, with regard to musculoskeletal disorders, we added the last sentence below to the discussion (p. 14, line 27):

(.)Moreover, although a broad range of chronic diseases has been investigated in this study, several conditions that are associated with unemployment have not been included, such as back pain or musculoskeletal disorders. In the current study, it was not possible to identify these chronic conditions by the use of medication data. For instance, medication that is prescribed for back pain includes over the counter pain killers such as paracetamol or ibuprofen. However, since pain killers are used for various forms of bodily pain and no information was available regarding the reason of prescription, it was not possible to identify these health conditions. Nevertheless, it is possible that inflammatory conditions include musculoskeletal problems, as NSAID's are a common treatment. (26)

Comment 9

All tables and figures should be at the end of the main text.

Response to comment 9

The guidelines of the journal of BMJ open allowed us to place the Tables in text.

Comment 10

Table 1. I suggest to add pvalues for differences between the two groups prevalence.

Response to comment 10

We thank the reviewer for the suggestion. Due to the large study population, we think that adding p-values to the table will not have additional value. This has been addressed in the discussion:

New text (Discussion, p.14, line 15):

Another strength of the current study is that our register-based data capture the whole Dutch population and therefore the data provide statistical power to investigate age-specific prevalences. This facilitates precise estimations of associations between health and employment.

Comment 11

Table 2. I suggest to place the columns with prevalence for 1,2 and >= chronic conditions in Table 1 (descriptive).

Response to comment 11

We have included the prevalence for the number of chronic diseases to Table 1.

Comment 12

Supplementary Table 2: The first 4 rows are single chronic diseases but no multimorbidity as the title of the table states.

Response to comment 12

We agree with this suggestion and think the following title is more appropriate (Results, p. 11):
The association of sociodemographic characteristics with the number of chronic diseases in the total population (n=5,074,227).

Comment 13

No ethical issues are addressed.

Response to comment 13

We have shortly addressed ethical issues in the methods:

New text (Methods, p. 5, line 7):

Register data covering information on all Dutch residents in 2016 were used. Statistics Netherlands provided individual-level databases on demographics, education, labor market status and prescribed medication. All Dutch residents were pseudonymized using a personal unique number. Data registries were linked at the individual level using these pseudonymized numbers. *No informed consent was needed for this study since authorized research institutes in the Netherlands are by law allowed to use pseudonymized register-based data for research purposes.*

. van Rijn RM, Robroek SJ, Brouwer S, Burdorf A. Influence of poor health on exit from paid employment: a systematic review. *Occup Environ Med.* 2014;71(4):295-301.

VERSION 2 – REVIEW

REVIEWER	Finn Breinholt Larsen DEFACTUM, Public Health and Health Services Research, Denmark
REVIEW RETURNED	07-May-2020

GENERAL COMMENTS	This paper fills a void in the literature on chronic illness/multimorbidity and affiliation with the labor market. The research is based on strong register data combining employment data with prescription data. The study has been carefully reviewed and comments from reviewers have been adequately addressed. I find the paper suitable for publication. NB. Page 9, line 25-26: "Supplementary Figure 1" should be replaced by "Supplementary Table 1"
--

REVIEWER	Josue Almansa
-----------------	---------------

	Department of Health Sciences, University Medical Center Groningen (UMCG) The Netherlands
REVIEW RETURNED	08-May-2020

GENERAL COMMENTS	Authors have addressed with detail all issues and comments from my first review. I would only like to mention two misspellings: In page 9 around line 23: "highestamong", two words are stuck together. In page 15 around line 13: I guess you mean "a decline" instead of "disability benefits, la decline".
---